# Epitranscriptional m⁶A modification of rRNA negatively impacts translation and host colonization in *Staphylococcus aureus*

Kathryn E. Shields[1], David Ranava[2], Yongjun Tan[3], Dapeng Zhang[3,4], Mee-Ngan F. Yap[1,2]*

1 Department of Biochemistry and Molecular Biology, Saint Louis University School of Medicine, Saint Louis, Missouri, United States of America, 2 Department of Microbiology-Immunology, Northwestern University Feinberg School of Medicine, Chicago, Illinois, United States of America, 3 Department of Biology, College of Arts and Sciences, Saint Louis University, St. Louis, Missouri, United States of America, 4 Program of Bioinformatics and Computational Biology, College of Arts and Sciences, St. Louis, Missouri, United States of America

* frances.yap@northwestern.edu

**Data Availability Statement:** The Ribo-seq and mRNA-seq data reported in this paper have been deposited in the NCBI Gene Expression Omnibus (GEO) database with accession number

## Abstract

Macrolides, lincosamides, and streptogramin B (MLS) are structurally distinct molecules that are among the safest antibiotics for prophylactic use and for the treatment of bacterial infections. The family of **e**rythromycin **r**esistance **m**ethyltransferases (Erm) invariantly install either one or two methyl groups onto the $N^{6,6}$-adenosine of 2058 nucleotide (m⁶A2058) of the bacterial 23S rRNA, leading to bacterial cross-resistance to all MLS antibiotics. Despite extensive structural studies on the mechanism of Erm-mediated MLS resistance, how the m⁶A epitranscriptomic mark affects ribosome function and bacterial physiology is not well understood. Here, we show that *Staphylococcus aureus* cells harboring m⁶A2058 ribosomes are outcompeted by cells carrying unmodified ribosomes during infections and are severely impaired in colonization in the absence of an unmodified counterpart. The competitive advantage of m⁶A2058 ribosomes is manifested only upon antibiotic challenge. Using ribosome profiling (Ribo-Seq) and a dual-fluorescence reporter to measure ribosome occupancy and translational fidelity, we found that specific genes involved in host interactions, metabolism, and information processing are disproportionally deregulated in mRNA translation. This dysregulation is linked to a substantial reduction in translational capacity and fidelity in m⁶A2058 ribosomes. These findings point to a general "inefficient translation" mechanism of trade-offs associated with multidrug-resistant ribosomes.

## Author summary

The Erm rRNA methyltransferases are widespread among nosocomial and commensal Gram-negative and Gram-positive bacteria. Upon exposure to sublethal doses of macrolide antibiotics, the expression of *erm* genes is upregulated, and Erm enzymes exquisitely methylate the universally conserved A2058 nucleotide of 23S rRNA in the bacterial ribosomes (m⁶A2058 ribosomes). The m⁶A mark prevents binding of all three clinically

GSE168265. All other relevant data are within the manuscript and its Supporting Information files.

**Funding:** This study was supported by the National Institutes of Health grants R01AI150986 (to MNFY) and R01GM121359 (to MNFY) and Department of Defense W81XWH-18-1-0122 (to MNFY), and in part by the PEW Charitable Trusts Grant #2920 (to MNFY) and the Edward Mallinckrodt Jr. Foundation (to MNFY). TEM analysis was performed at the Northwestern University Center for Advanced Microscopy generously supported by NCI CCSG P30 CA060553. The funders had no role in study design, data collection and analysis, decision to publish, or preparation of the manuscript.

**Competing interests:** The authors have declared that no competing interests exist.

important classes of antibiotics (macrolides, lincosamides and streptogramin B (MLS)). While the mechanism of MLS resistance has been studied extensively, how the methylation affects protein synthesis and bacterial fitness is less well understood. We show that human pathogenic *Staphylococcus aureus* cells bearing m$^6$A2058 ribosomes are defective in colonizing animals and have inferior competitive fitness to cells carrying unmodified ribosomes. Global and targeted translation analyses revealed that m$^6$A2058 ribosomes exhibit decreased overall translational efficiency. Our data add to a growing list of examples linking fitness cost to the acquisition of resistance genes.

## Introduction

Adenosine methylation of RNA is one of the most abundant chemical modifications in all living organisms. Adenosine can be methylated at the nitrogen and carbon atoms or the 2'-O position of ribose in virtually all types of RNAs, including mRNAs, ribosomal RNAs (rRNAs), tRNAs and regulatory RNAs [1]. Installation of N$^6$-methyladenosine (m$^6$A) in RNA can impact translational efficiency, RNA stability, folding, export, processing, and RNA-protein or RNA-RNA interactions [2]. Approximately 25% of eukaryotic mRNAs contain m$^6$A, but m$^6$A has been detected in less than 0.3% of bacterial mRNAs [3]. In eukaryotic rRNA, m$^6$A modifications lead to an increase in the translation of a selective set of transcripts associated with the stress response and cell proliferation [4–8]. Methylated rRNA nucleotides in bacteria are generally found in clusters near the ribosomal active sites, on the surface of 16S rRNA, or partially buried in 23S rRNA, which reflects a particular stage by which methylation occurs during ribosome assembly [9, 10]. The loss of housekeeping rRNA methylation in bacteria is linked to slower rates and lower accuracy of translation, impaired responses to metabolites, and defective 17S rRNA processing [10–12].

Methylation of bacterial rRNA via the acquisition of specific RNA methyltransferase genes often gives rise to antibiotic resistance [13–16]. One such example is the dimethylation of the exocyclic N6 amine of a universally conserved A2058 nucleotide (*E. coli* numbering, hereafter referred to as m$^6$A 2058) in the 23S rRNA by the erythromycin-resistance methylase (Erm) family. m$^6$A2058 decreases the affinity of macrolide antibiotics for their binding site by precluding water-mediated interactions between the desoamine sugar of the macrolide and A2058 (Fig 1A)[17]. m$^6$A2058 not only confers cross-resistance against all three critically important classes of antibiotics (**m**acrolides, **l**incosamides, and **s**treptogramin B(**MLS**)) but also camouflages bacteria from recognition by the Toll-like receptors (TLRs), thereby evading the innate immune response and facilitating host infections [18–20]. The *erm* genes are widespread among nosocomial Gram-negative and Gram-positive bacteria [21–31]. Erm-based resistance often represents up to 50–98% of MLS-resistant isolates, surpassing MLS resistance caused by target site mutations, drug inactivation and efflux combined [28,32–34]. All *erm* genes are invariantly located downstream of a short ribosome stalling leader sequence in a two-gene operon (Fig 1B)[15,35]. Previous studies have shown that the expression of *erm* is upregulated when the macrolide-bound ribosome stalls at a specific site in the leader peptide preceding the cotranscribed *erm*. Ribosome stalling presumably activates Erm synthesis by altering the *erm* mRNA secondary structure, which otherwise occludes the *erm* Shine-Dalgarno (SD) sequence from translational initiation [35–39]. The ribosome stalling model is not the only mechanism of Erm upregulation [40,41], because mutations that abolish the translation of the leader peptide or delete the upstream regulatory region are prevalent in MLS-

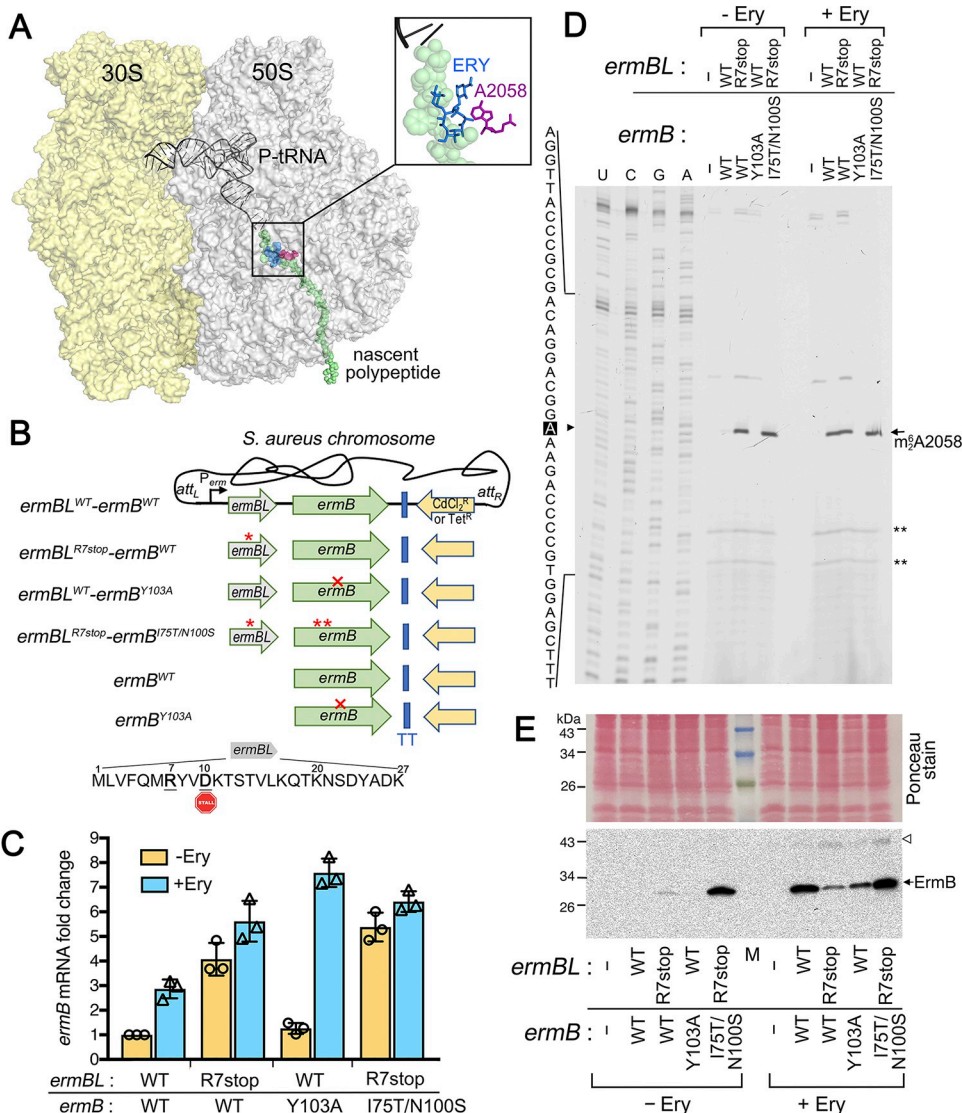

**Fig 1. Modulation of inducible and constitutive MLS resistance by the upstream regulatory sequence in *S. aureus*.** (**A**) A simplified structure of the bacterial 70S ribosome (PDB 3J5L [117]), with a modeled nascent peptide (green) showing the interaction between the A2058 nucleotide (magenta) and erythromycin (ERY, blue) inside the ribosome exit tunnel. P-site tRNA is shown to localize to the peptidyl transferase center (PTC). (**B**) Reconstruction of a single copy of *ermBL-ermB* or *ermB* alone on a neutral chromosomal *att* site in MLS-sensitive *S. aureus* USA300 JE2. The integrated *ermBL-ermB* or *ermB* is under the control of its native promoter and is marked with either a cadmium chloride resistance ($CdCl_2^R$) or a tetracycline resistance ($Tet^R$) cassette. Mutant alleles, including the premature ochre codon in position Arg-7 of *ermBL*$^{R7stop}$, catalytically inactive *ermB*$^{Y103A}$, and hyperactive *ermB*$^{I75T/N100S}$ are indicated by an asterisk or a crossmark. The ErmBL sequence containing the ribosome stalling motif VDK is shown. ERY-bound ribosomes arrest at D10 in the P-site of the PTC (stop sign)[40]. TT, transcriptional terminator. (**C**) RT-qPCR confirming that *ermB* expression is inducible by sublethal doses of ERY (1 μg/mL, 60 min) in the context of both translationally functional and defective *ermBL*. The relative expression of *ermB* was normalized by *ermBL*$^{WT}$-*ermB*$^{WT}$ using *polC* as the internal reference gene. Error bars indicate ±SD. (**D**) Primer extension showing the magnitude of A2058 methylation in the 23S rRNA. The reverse transcriptase halts one nucleotide before the methylated site and produces a truncated cDNA that is manifested by a strong signal at A2058 analyzed on a denaturing polyacrylamide gel. ** serves as an internal reference for equal rRNA template. Each reaction contains 250 ng of rRNA input. (**E**) Western blots showing the steady-state protein levels of ErmB in the presence and absence of ERY. Samples were collected in parallel with the RNA extraction in Panels C-D. An open arrow marks the non-specific antibody cross-reaction. M, protein marker. A minimum of 3 independent biological replicates were performed.

resistant clinical isolates [42–52], and in some cases Erm overexpression is associated with mRNA stabilization independently of the ribosome stalling mechanism [40].

While the ribosome stalling-dependent regulation of *erm* has been studied extensively [35,38–41], the cellular impact of m$^6$A2058 ribosomes on bacterial physiology and pathogenesis is not well understood. In this work, we use the *ermBL-ermB* operon ("L" indicates leader sequence) from a methicillin-resistant *Staphylococcus aureus* (MRSA) as our model system to investigate the effects of m$^6$A2058 on global translation. Our findings link the in vivo fitness loss of the m$^6$A2058-bearing strain to the downregulation of major virulence genes and demonstrate that a single m$^6$A modification in the ribosome can significantly alter transcript levels and translational output.

## Results

### The *ermBL* leader is indispensable for the regulation of MLS resistance phenotypes

CM05 is a methicillin-resistant *S. aureus* that carries chromosomally encoded Cfr rRNA methyltransferase, the macrolide efflux pump proteins MefA and MsrA, and the genetically linked *ermBL-ermB*, collectively conferring resistance to all the clinically relevant antibiotics that target the 50S ribosomal subunit [53]. To simplify interpreting the resistance phenotype, we reconstituted a single copy *ermBL-ermB* in the MLS-sensitive MRSA strain USA300 JE2 by integrating the native promoter-containing operon into a neutral chromosomal *att* site. The integration system has been widely used for genetic complementation in *S. aureus* [54–57]. The USA300 pulsotype was chosen because this lineage is the primary source of MRSA infections in North America and often carries *erm*-containing plasmids [32,58]. Using site-directed mutagenesis, we introduced a nonsense mutation at the R7 position of the ErmBL leader peptide (*ermBL*$^{R7stop}$-*ermB*$^{WT}$). This mutation abolishes translational stalling due to premature termination before the $^9$VDK$^{11}$ ribosome stalling motif and increases *ermB* transcripts by 4-fold compared to *ermBL*$^{WT}$-*ermB*$^{WT}$, consistent with previous findings (Fig 1C)[40]. The mechanism of mRNA stabilization upon abrogation of ribosome stalling and erythromycin (a macrolide) treatment remains unclear [40,59]. The catalytically inactive mutant *ermBL*$^{WT}$-*ermB*$^{Y103A}$ served as a negative control for methyltransferase activity and an in vitro evolved *ermBL*$^{R7stop}$-*ermB*$^{I75T/N100S}$ variant served as a positive control for hypermethylation. We confirmed by primer extension that in the absence of sublethal erythromycin inducer, the *ermBL*$^{R7stop}$-*ermB*$^{WT}$ strain is constitutively methylated at A2058, and the WT strain has undetectable level of methylation, whereas *ermB*(Y103A) is completely unmodified (Fig 1D), in agreement with the steady-state ErmB protein levels detected by Western blots (Fig 1E). How The ErmB$^{I75T/N100S}$ variant enhances protein stability remains to be investigated. In the presence of erythromycin, the degree of m$^6$A2058 methylation in the *ermBL*$^{WT}$-*ermB*$^{WT}$ strain was comparable to that in the constitutively expressed *ermBL*$^{R7stop}$-*ermB*$^{WT}$ (Fig 1D). By applying the principle of dideoxy sequencing in a primer extension assay [60], we estimated that 25%-51% of the ribosomes in the chromosomally-encoded ErmB strains are dimethylated at A2058 (S2 Fig).

The hyper-MLS resistance phenotype of the *ermBL*$^{R7stop}$-*ermB*$^{WT}$ strain is corroborated by the minimum inhibitory concentration (MIC) of various antibiotics. Of note, *ermBL*$^{WT}$-*ermB*$^{WT}$ exhibits modest resistance to macrolides (Ery, AZI), lincosamide (CLN) and the third-generation macrolide-ketolide (SOL) whereas *ermBL*$^{R7stop}$ variants are hyperresistant to all tested inhibitors of 50S ribosomal subunit (Table 1). ErmB-mediated resistance is specific to MLS antibiotics because neither the *erm*-deficient nor *erm*-proficient S. *aureus* strains demonstrated any alteration in sensitivity to antibiotics that target other cellular targets, except

**Table 1. Minimum inhibitory concentration (MIC, µg/mL) of different antibiotics against *S. aureus* strains bearing the A2058 methylated- and unmethylated-ribosomes.** MICs were determined by E-test on Muller Hinton Agar plates in biological triplicates per strain per antibiotic type.

| Cellular targets | Abx[1] | Parental JE2 | JE2 *att*:pJC1111:*ermBL-ermB*-CdCl$_2^R$ | | | | JE2 *att*:pJC1306:*ermBL-ermB*-Tet$^R$ | |
| --- | --- | --- | --- | --- | --- | --- | --- | --- |
| | | — | *ermBL*$^{WT}$-*ermB*$^{WT}$ | *ermBL*$^{R7stop}$-*ermB*$^{WT}$ | *ermBL*$^{WT}$-*ermB*$^{Y103A}$ | *ermBL*$^{R7Stop}$-*ermB*$^{I75T/N100S}$ | *ermBL*$^{WT}$-*ermB*$^{WT}$ | *ermBL*$^{R7stop}$-*ermB*$^{WT}$ |
| 50S ribosomal subunit | ERY | 0.19 | 12 | >256 | 0.095 | >256 | 12 | >256 |
| | AZI | 0.38 | 48 | >256 | 0.125 | >256 | 64 | >256 |
| | CLN | 0.032 | 1.5 | >256 | 0.016 | >256 | 1.5 | >256 |
| | SOL | 0.064 | 2 | 6 | 0.125 | >32 | 1.5 | 6 |
| 30S ribosomal subunit | KAN | 2 | 1.5–2 | 1.5 | 1.5 | 1.5 | 1.5 | 1.0 |
| | TOB | 0.38 | 0.38 | 0.38 | 0.38 | 0.38 | 0.25 | 0.25 |
| | STP | 3 | 3 | 3 | 3 | 3 | 3 | 3 |
| Cell wall/ membrane | DAP | 1 | 0.5 | 0.125 | 0.25 | 0.125 | 0.125 | 0.25 |
| | VAN | 1.5 | 1.5 | 1.5 | 1.5 | 1.5 | 1.5 | 1.5 |
| | FOS | 1.5 | 1.5 | 1 | 1.5 | 1 | 1.5 | 1 |
| RNA polymerase | RIF | 0.006 | 0.006 | 0.006 | 0.006 | 0.006 | 0.006 | 0.006 |
| Dihydrofolate reductase | TMP | 0.064 | 0.064 | 0.047 | 0.064 | 0.047 | 0.064 | 0.047 |
| DNA Topoisomerase | NAL | 96 | 96 | 96 | 96 | 96 | 96 | 96 |
| | CIP | >32 | 24 | 16 | >32 | 16 | nd[2] | nd[2] |

[1] Antibiotics (Abx); Erythromycin (ERY); Azithromycin (AZI); Clindamycin (CLN); Solithromycin (SOL); Kanamycin (KAN); Tobramycin (TOB); Fosfomycin (FOS); Streptomycin (STP); Daptomycin (DAP); Vancomycin (VAN); Rifampicin (RIF); Trimethoprim (TMP); Nalidixic acid (NAL); Ciprofloxacin (CIP).

[2] nd, not determined.

for *ermBL*$^{R7stop}$ variants that are moderately more susceptible to ciprofloxacin than the parental strains (Table 1), implying possible collateral sensitivity because of ErmB overexpression.

We found that the inclusion of the *ermBL* coding region is critical because strains without it failed to recapitulate the MLS resistance traits despite being under the control of its native promoter (Fig 1B and S1 Table), reinforcing the important regulatory function of the *ermBL* sequence.

## Constitutive m$^6$A2058 modification of ribosomes severely attenuates competitive *S. aureus* colonization

To ensure reproducibility and to prevent secondary mutations introduced during genetic manipulations, we created two independent sets of *ermBL-ermB* integrated strains, each carrying a CdCl$_2$ or a tetracycline (Tet) resistance cassette. The CdCl$_2$ and Tet cassettes did not affect MLS resistance profiles (Table 1). These selective markers allowed us to compare the competitive fitness of inducible and constitutive *ermB* expression in an in vitro competition and in vivo murine model of coinfections. We first performed in vitro competitions between pairs of inducible *ermBL*$^{WT}$-*ermB*$^{WT}$ and constitutive *ermBL*$^{R7stop}$-*ermB*$^{WT}$ strains in the presence and absence of 1.5× MIC erythromycin. Competitiveness was evaluated by enumeration of the total CFU of each strain after a 1:1 ratio of the initial inoculation. In the absence of erythromycin, *ermBL*$^{WT}$-*ermB*$^{WT}$ dominated over *ermBL*$^{R7stop}$-*ermB*$^{WT}$ by 50- to 100-fold in two independent combinations. Overall, the *ermBL*$^{R7stop}$-*ermB*$^{WT}$ strain exhibited poorer growth capacity even with an *ermBL*$^{R7stop}$-*ermB*$^{WT}$ kin (Fig 2A, top). Conversely, in the presence of erythromycin, inducible *ermBL*$^{WT}$-*ermB*$^{WT}$ grew poorly against the constitutive *ermBL*$^{R7stop}$-*ermB*$^{WT}$ (Fig 2A, bottom). In a murine model of sepsis, inducible *ermBL*$^{WT}$-*ermB*$^{WT}$

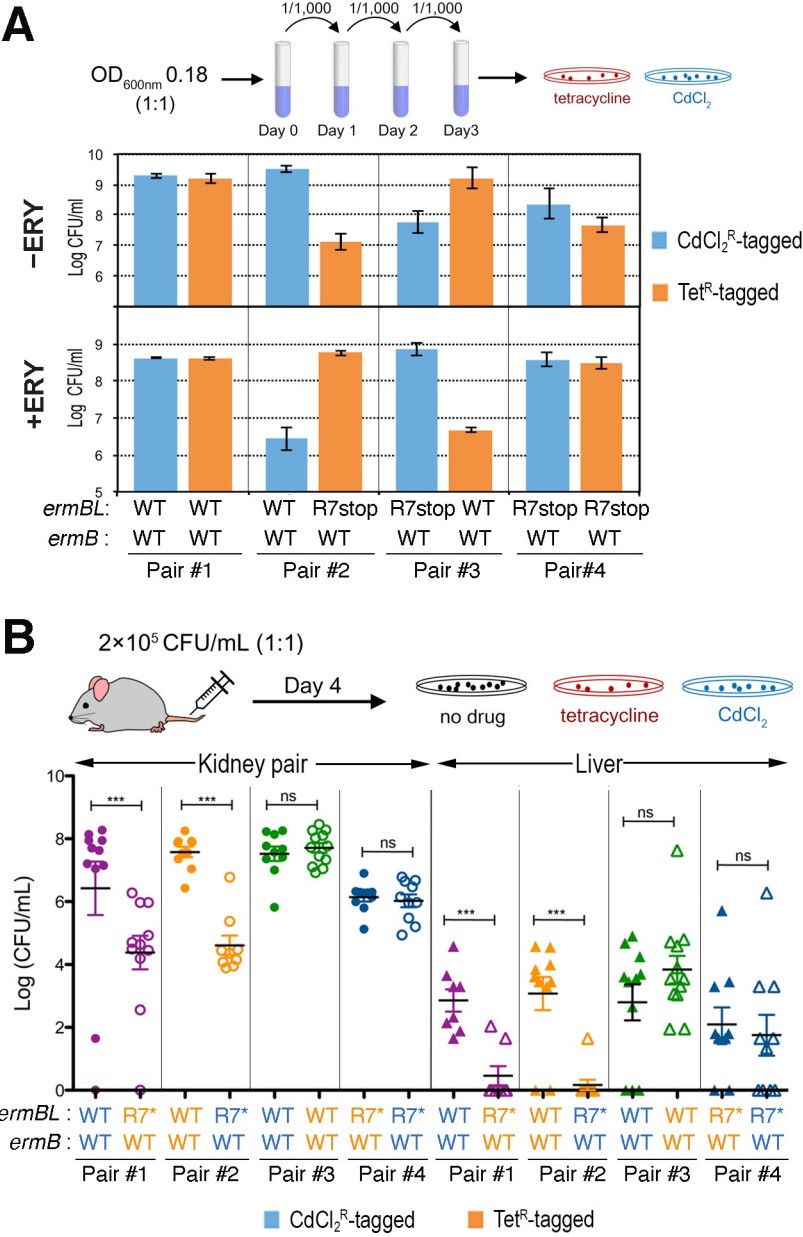

**Fig 2. In vitro and in vivo competition assays of *S. aureus* strains bearing ERY-inducible *ermBL*$^{WT}$-*ermB*$^{WT}$ and constitutively expressed *ermBL*$^{R7stop}$-*ermB*$^{WT}$.** Each strain was tagged with either CdCl₂ or a tetracycline selective marker. **(A)** (Top) The *ermBL*$^{R7stop}$-*ermB*$^{WT}$ strains were >100-fold less competitive than the *ermBL*$^{WT}$-*ermB*$^{WT}$ strains in tryptic soy broth without erythromycin (ERY, a macrolide). (Bottom) In the presence of 20 µg/mL ERY, the *ermBL*$^{R7stop}$-*ermB*$^{WT}$ strain outgrows the *ermBL*$^{WT}$-*ermB*$^{WT}$ strain. **(B)** The *ermBL(R7Stop)*-*ermB* strains were severely attenuated in a murine sepsis model (N = 12 mice/group). Bacterial burden was measured in livers and kidneys 4 days post inoculation. Each data point is the mean value ±SEM. Statistical significance was determined by 1-way ANOVA with Tukey's test (***, p<0.005; ns, not significant). An asterisk in R7 indicates *ermBL*$^{R7stop}$ and the CdCl₂-tag and a Tet-tag are color-coded in blue and orange, respectively.

dominated the kidney and liver burdens on Day 4 over *ermBL*$^{R7stop}$-*ermB*$^{WT}$ by three orders of magnitude. Consistent with the in vitro findings, the isogenic *ermBL*$^{R7stop}$-*ermB*$^{WT}$ pair had a much lower bacterial load than the isogenic *ermBL*$^{WT}$-*ermB*$^{WT}$ pair in the absence of antibiotic selection pressure (Fig 2B). That inducible *ermB* expression has a substantial colonization

advantage over the constitutive *ermB* expression in vivo suggests that hypermethylation of ribosomal A2058 imposes a significant fitness cost, possibly due to the reprogramming of gene expression.

## Genome-wide ribosome profiling (Ribo-seq) reveals altered mRNA levels and decoding of a subset of virulence-associated genes

Morphological changes in pathogenic bacteria are hallmarks of altered colonization and virulence properties [61]. The reduced colonization competitiveness of the *ermBL*^R7stoP-*ermB*^WT strain may have altered cell shape. However, no detectable changes in cell morphology and size were observed in all ErmB-deficient and ErmB-overproducing strains using epifluorescence and transmission electron microscopic analysis (S1 Fig), suggesting that $m^6$A2058 ribosomes may only affect specific cellular pathways. To investigate the mechanism underlying the fitness loss of ribosome methylation, we carried out deep sequencing-based Ribo-seq in parallel with total mRNA-seq to capture a global snapshot of ribosome density and position on template mRNAs and changes in the whole transcriptome (Fig 3A). A high number of ribosome-protected footprints (RPFs) mapping to the transcriptome (mRNA-seq) at a unique position or being distributed across the transcript is indicative of ribosome stalling or active translation (indicated by translation efficiency (TE)), respectively. Our initial attempt to construct RPF and mRNA libraries from stationary phase cells produced variable results. We opted for exponentially grown *ermBL*^R7stoP-*ermB*^WT and *ermBL*^WT-*ermB*^Y103A cells, which represent strains with constitutively expressed ErmB and the catalytically inactive ErmB(Y103A) mutant, respectively (Fig 1D and 1E). Because of its known artifacts [62,63], the translation inhibitor chloramphenicol was excluded from cell treatment and sample harvest. No observable ribosome biogenesis defects, e.g., aberrant 30S:50S ratio and ribosomal intermediate, were observed in these strains based on the ribosome sedimentation profiles, but *ermBL*^R7stoP-*ermB*^WT had a modest accumulation of the translationally silent 100S ribosomes, which are known to be devoid of mRNA and tRNA (S3 Fig) [64,65], providing the first hint that *ermBL*^R7stoP-*ermB*^WT may have lower translational efficiency.

On average, we obtained more than 4 million uniquely mapped reads of mRNAs and RPFs, and the independent biological replicates were highly reproducible (S2 Table). Translational and transcriptional attenuation mediated by the 5' *cis* regulatory elements are understudied in Firmicutes with the exception of T-box mechanism and riboswitches [66], and a handful examples of antibiotic-sensing arrest peptides [35]. The well-characterized ribosome stalling peptides SecM, MifM, TnaC, VemP, SpeFL are absent in *S. aureus* [67,68]. Nevertheless, we detected ribosome pausing of known upstream leader peptide in the *S. aureus ilv-leu* operon encoding branch-chained amino acids biosynthesis genes [69] and changes in ribosome density at the known sites of programmed ribosomal frameshifting within the *prfB* gene (S4 Fig). These observations offer high confidence for our translatome data. To better discern translational changes from changes in mRNA levels, we calculated the TE in the *ermBL*^R7stoP-*ermB*^WT($m^6$A2058) and *ermBL*^WT-*ermB*^Y103A (A2058) by dividing RPFs by mRNA-seq density. Local accumulation of ribosomes at specific codon positions of mRNA followed by sparse downstream ribosome density is indicative of ribosome stalling. Such a signature was not observed in either strain except for *ilv-leu* leader peptide reference described above. Plotting the mRNA and RPF ratios provided information about the changes in transcript levels and ribosome occupancy between strains, allowing changes in mRNA levels to be distinguished from alterations in mRNA translation (Fig 3B). We found that a large fraction of genes involved in metabolism and information processing and many well-characterized virulence genes were concordantly downregulated at the transcript and translation levels, including

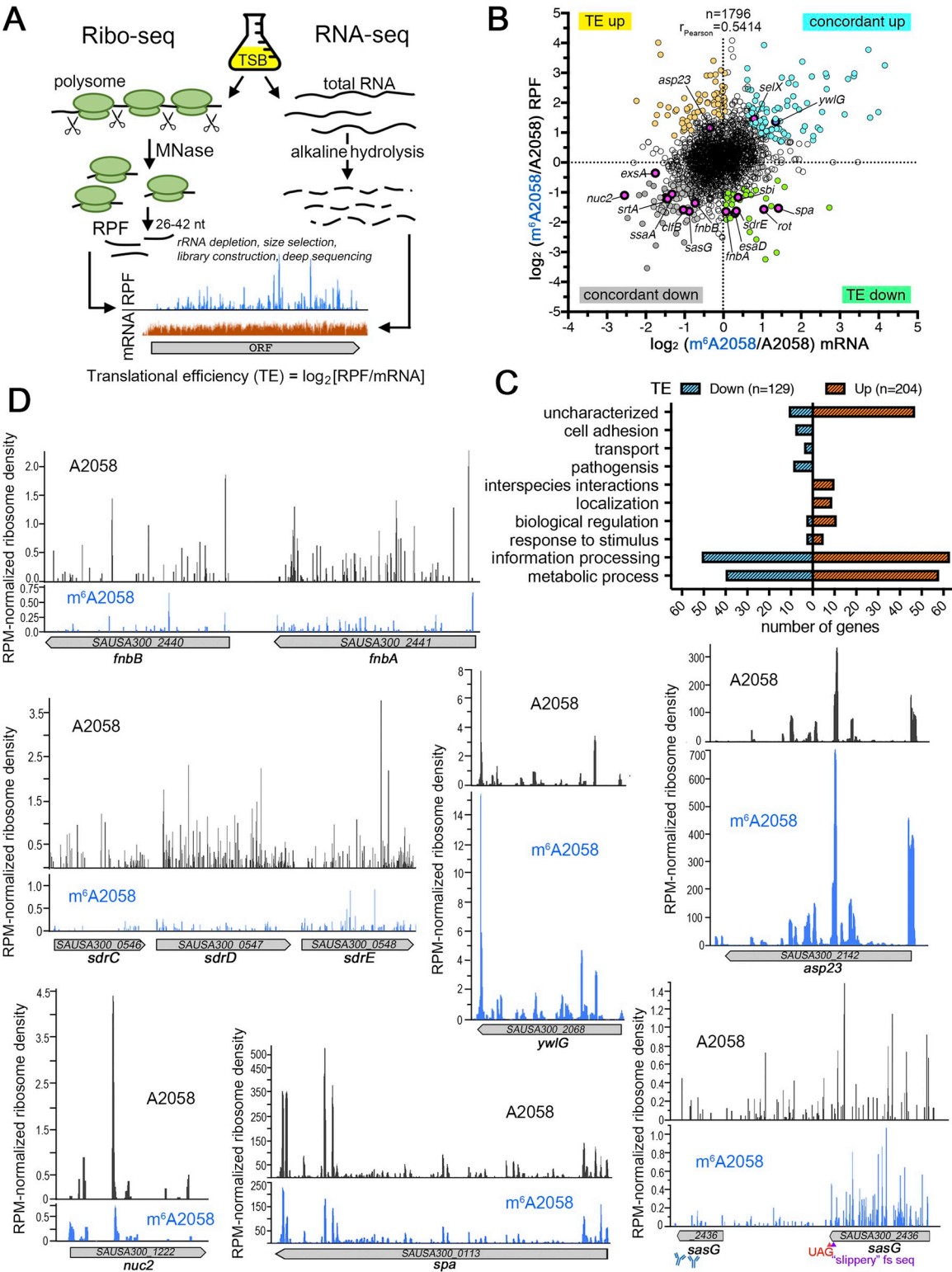

**Fig 3. Effects of m⁶A2058 modification on the global translatome.** (**A**) Experimental workflow of Ribo-seq and total mRNA-seq. Translational efficiency (TE) was calculated as the relative number of ribosome footprints to mRNA-seq reads in log$_2$ ratios. A higher TE value represents a greater potential of mRNA for translation. (**B**) The log$_2$ fold change in Ribo-seq or mRNA levels are plotted for all genes expressed under the tested conditions. The mRNA levels were calculated in reads per kilobase per million mapped reads (RPKM). Representative virulence-associated genes are marked. (**C**) Comparison of TEs between the m⁶A2058 strain and unmodified strain

reveals the inventory of differentially translated mRNAs. GO analysis showed enrichment in the biological processes of gene expression, metabolic processes, and pathogenesis. **(D)** Representative ribosome density plots showing changes in ribosome occupancy along the gene body. The normalized reads per million mapped reads (RPM, y-axis) correspond to the average ribosome density across the most abundantly translated ORFs (>50 reads). Unexpected ribosome occupancy is detected in the 3'-end of *sasG* mRNA, suggesting low levels of mRNA translation downstream of the UAG codon. The potential frameshift-prone "slippery" sequence, the premature stop codon, and the epitope of anti-SasG are indicated.

many that are required for cell adhesion, such as *fnbAB*, the *sdr* operon, *nuc2*, *sasG* and *spa* (Fig 3B–3D and S1 Dataset). Intriguingly, the SasG protein is known to be truncated and non-functional in USA300 strains due to a single nucleotide deletion that introduces premature stop codons at its C-terminus [70]. Instead, we observed ribosome occupancy downstream of the UAG codon in both unmodified A2058 and m⁶A2058 strains, albeit at a much lower level for m⁶A2058 ribosomes (Fig 3E), suggesting the possible occurrence of stop codon read-through or ribosomal frameshift events. We found that the number of upregulated genes in TE (n = 204) exceeds genes that are downregulated in TE (n = 129) (Fig 3C). How m⁶A2058 modification increases transcript levels and TE remains to be investigated.

## m⁶A2058 ribosomes reduce translational output

Our genome-wide findings demonstrate that a reduction in cell surface/membrane localized virulence factors could partially account for the loss of colonization fitness in the *ermBL*^R7stop^-*ermB*^WT^(m⁶A2058) strain (Fig 2B). To validate the effect of m⁶A2058 on the protein yield from transcripts that were identified from Ribo-seq, we performed Western blotting on selected virulence factors, SdrE [71], Nuc2 [72], Asp23 [73], SpA [74], YwlG [75] and SasG [76], to measure steady-state protein levels in both ErmB-proficient and ErmB-deficient *S. aureus*. Our validation experiments were limited by the availability of antibodies. Genetic knockouts or a deletion mutant of the SigB positive regulator were included to warrant antibody specificity. Immunoblotting confirmed the downregulation (SdrE, Nuc2, SpA, SasG) and upregulation (Asp 23, YwlG) trends of the m⁶A2058 strains (Fig 4). Partial Nuc2 degradation in the parental *ermBL*⁻-*ermB*⁻ strain might contribute to its lower level, albeit the reason is unclear. Of note, the production of the full-length copy of SasG in the catalytically inactive ErmB(Y103A) strain was detected using the anti-SasG raised from a C-terminal epitope of SasG (NH₂-ESTQKGLIFSSIIGIAGLMLLAR), consistent with the ribosome density plot showing the potential translational recoding upstream of the UAG codon to produce low levels of SasG (Figs 3D and 4).

## m⁶A2058 reduces translational efficiency and fidelity

Bacterial translational errors range between $10^{-3}$–$10^{-4}$ per codon [77], and translational efficiency is linked to mRNA stability and transcription [78]. We posit that m⁶A2058 ribosomes may be prone to translational inaccuracies, including frameshifts and readthrough, due to inefficient translational elongation. To examine frameshifts and readthrough, we transferred a well-established *mCherry-yfp* dual fluorescence reporter system to a stable *S. aureus* plasmid under the control of a constitutive *hpf* promoter [79,80]. To eliminate transcriptional noise, these *mCherry* and *yfp* genes were cotranscribed and translated from the *mCherry* start codon into a single mCherry-YFP polypeptide in the control "WT reporter". At the end of the "error reporter", mCherry was introduced with either a stop codon or a +1 or -1 frameshift [81]. Readthrough of the stop codon or frameshifting produced a fusion that yields YFP signals, while accurate translation produced only mCherry in the "error reporter" (Fig 5A). The degree of translation was calculated as the YFP/mCherry ratio relative to the 'WT reporter'

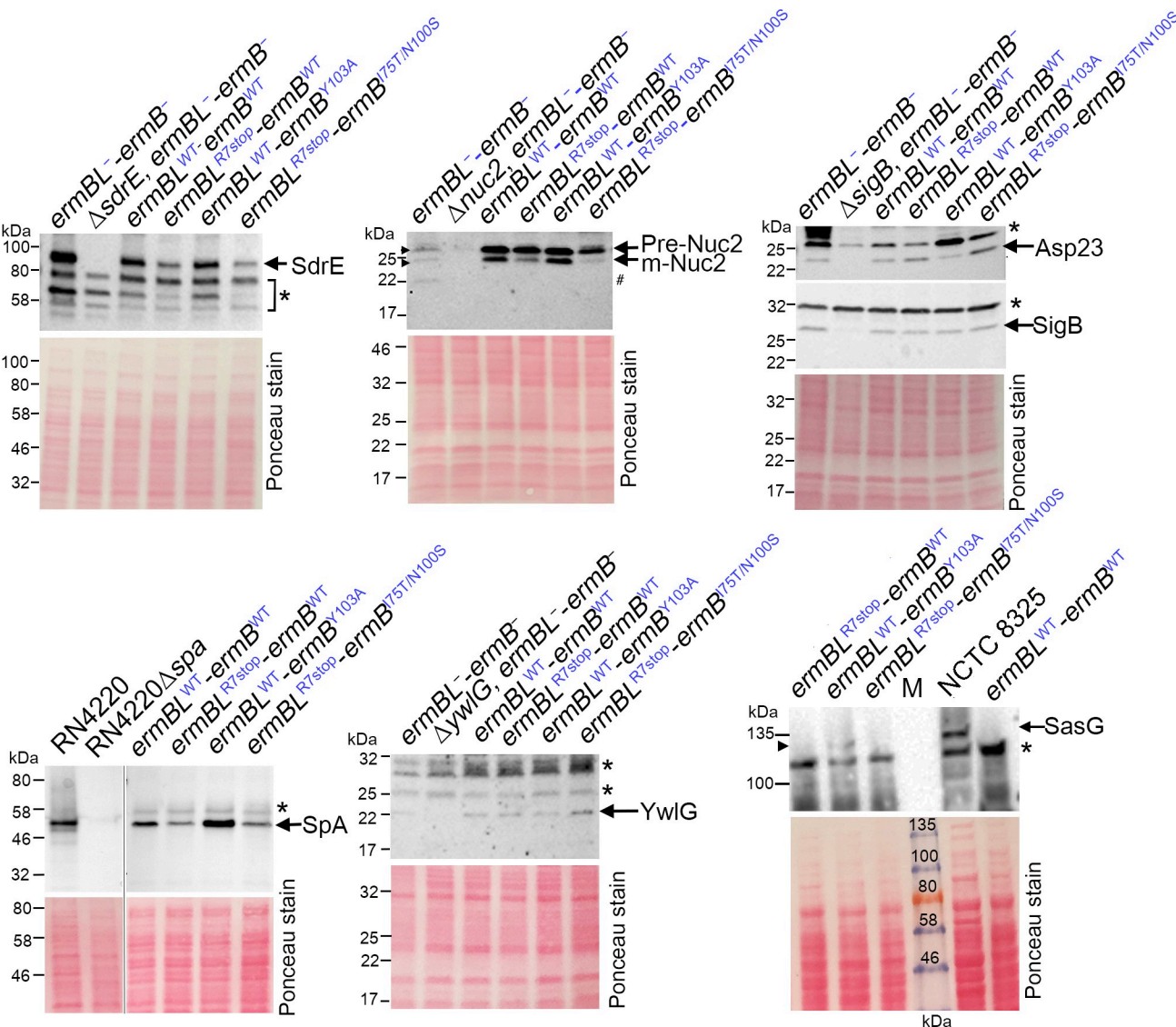

**Fig 4. Comparison of steady-state protein levels between the m⁶A2058 ribosome-bearing strain and its unmodified counterpart by Western blotting.** Cell lysates were prepared from strains grown under the conditions shown in Fig 3. Ponceau S staining of the membranes prior to immunoblotting served as the loading control. *S. aureus* NCTC 8325 encodes a full-length SasG and served as a control. The specificity of antibodies was validated by a gene knockout or a deletion mutant of a positive regulator, i.e., SigB. Both precursor (pre-Nuc2) and secreted mature forms (m-Nuc2) of Nuc2 were detected in the total cell lysates. Possible Nuc2 protein degradation (#) is observed in the *ermBL⁻-ermB⁻* strain, resulting in weaker pre-Nuc2 and m-Nuc2 signals. Each lane corresponds to 0.1–0.2 A₂₈₀ units of total proteins. Proteins were resolved on a 4–20% TGX SDS-PAGE gel and probed with the indicated antibodies. An asterisk indicates nonspecific cross-reaction of the antibodies.

normalized by cell number and compared between the *ermBL*^WT^-*ermB*^Y103A^ and *ermBL*^R7stop^-*ermB*^WT^ strains. A significant reduction in WT mCherry-YFP was observed in the m⁶A2058 strain, suggesting poorer translational elongation than in the unmethylated A2058 counterpart. Additionally, the m⁶A2058 strain exhibited lower UAG readthrough but slightly higher +1 ribosomal frameshifting than the A2058 strain (Fig 5B). A survey of start and stop codon usage in *S. aureus* USA300 revealed that UAA is used in >70% of the ORFs (Fig 5C) and that UAA appears to halt translation more effectively in all strains (Fig 5B), supporting a general conjecture that evolutionary pressure drives more genes to develop strong termination codons

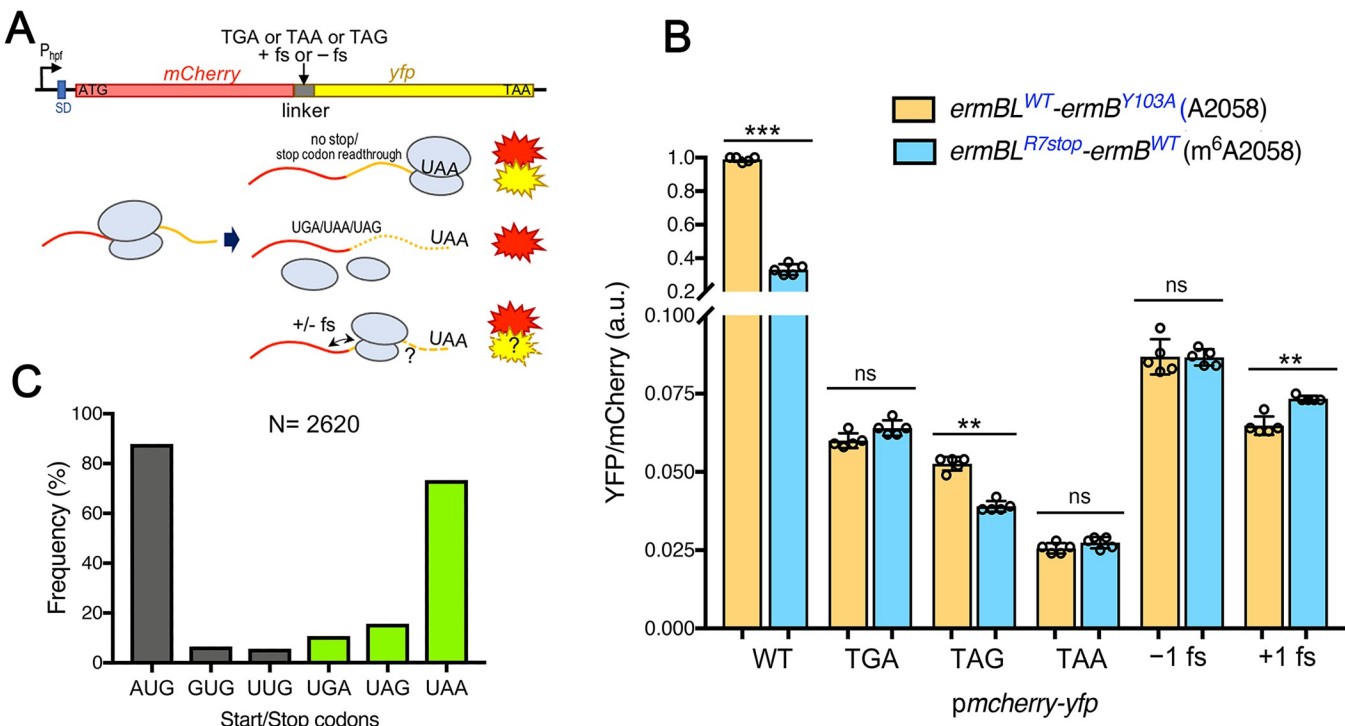

**Fig 5. Measurement of translational output and accuracy in m⁶A2058 ribosomes using a dual-fluorescence reporter.** (A) Construction of an *S. aureus*-specific *mCherry-yfp* reporter under the control of a constitutive *hpf* promoter that was adapted from a previously described *E. coli* system [81]. Nonsense codons and frameshift insertions were introduced between the *mCherry* and *yfp* to evaluate stop codon readthrough and ribosomal frameshifting (fs). (B) The m⁶A2058 ribosome-bearing strain exhibits poorer translational capacity and UAG bypassing but slightly higher +1 fs. Error bars represent ±SD (n = 5). P values were determined using unpaired t tests. ***p< 0.005, **p< 0.01; ns, not significant. (C) Genomic surveillance of start and stop codon usage in *S. aureus* USA300. AUG and UAA are the most frequently used start and stop codons, respectively.

that resist readthrough, whereas UGA may be more prone to readthrough [81,82]. We found that translational readthrough at the amber codon UAG was significantly lower in the m⁶A2058 strain, offering a plausible explanation for the low translational elongation downstream of the premature UAG codon in the *sasG* mRNA (Fig 3).

## Discussion

*S. aureus* is the second leading bacterial pathogen in worldwide deaths associated with antimicrobial resistance [83,84]. The usefulness of MLS antibiotics has rapidly eroded due to the widespread transmission of *erm* genes in *S. aureus* and other bacterial species [32]. *erm* genes can be either constitutively expressed or inducibly upregulated by a sublethal concentration of macrolides [35,85]. This inducible mechanism allows tunable expression of Erm, which is necessary for survival in the presence of MLS antibiotics [60]. In this work, we found that constitutive expression of ErmB because of disruption in the ErmBL regulatory region leads to a significant alteration of the global translatome. Although some translatomic changes could be due to an indirect effect of methylated ribosomes on the mRNA, such as potential decoupling of transcription-translation and translation-dependent mRNA decay [78,86], many key virulence genes are downregulated in *S. aureus* carrying the m⁶A2058 ribosomes concomitantly with an attenuation of host colonization (Figs 2 and 3). Our complementary reporter assays further showed that m⁶A2058 ribosomes are less efficient in translation and have altered translational recoding activity. These results demonstrate that a single posttranscriptional

modification of the rRNA can create a pool of "alternative ribosomes" and have a profound consequence on translation and infection outcomes.

"Specialized ribosomes" has been a concept of intense debate in both prokaryotic and eukaryotic systems [87,88]. In bacteria, preferential mRNA translation has been reported for alternative mycobacterial ribosomes carrying an S18 ribosomal paralog [89]. Nucleotide variations in the bacterial and mammalian rRNAs are known to contribute to stress adaptation and tissue specificity, respectively [90–92], and many RNA modifications are associated with human diseases and antimicrobial resistance [93]. A bacterial $m^2A$ RNA methyltransferase (RlmN) with dual targeting of rRNA and tRNA has been shown to regulate the translation of stress-related transcripts in response to reactive oxygen species [94]. Our Ribo-seq analyses did not detect unique codon occupancy or signatures of aberrant translation initiation and ribosome stalling in the mRNAs that are dysregulated in the $m^6A2058$ ribosomes (Fig 3). Due to the difficulty of obtaining a homogenous population of $m^6A2058$ ribosomes for in-depth biochemical investigations, our current study does not reveal the mechanism underlying the altered translational features of $m^6A2058$ ribosomes. A2058 is located at the constricted region of the ribosome tunnel ~20 Å away from the peptidyl transferase center (PTC) and acts as a functional site to monitor nascent peptide elements [95,96]. A2058 can tolerate some degree of mutational flexibility by preserving ~80% of PTC activity [97]. We posit that the addition of hydrophobic methyl groups may perturb the interaction between the growing nascent polypeptide and the ribosome tunnel, thereby reducing the translational elongation of specific mRNAs. Sequence context and mRNA secondary structure may shape the expression levels of individual genes [98]. However, none of the known proline-rich and arginine/lysine-rich ribosome stalling motifs [99–101] is associated with translational attenuation in the $m^6A2058$ strain. We initially speculate that the serine-aspartate dipeptide repeats (154–198 a.a. long) in the SdrCDE proteins are linked to reduced translation (Fig 3), but this sequence feature does not conform to those in other serine-aspartate repeat proteins, e.g., ClfA.

*S. aureus* USA300 carries a 1-nt deletion at the 1287th nucleotide of *sasG*, resulting in the premature termination of *sasG* translation at a UAG codon proximal to the deletion. Interestingly, the deletion is located within a classical "slippery" frameshift sequence (A-AAA-AAG, a dash indicates zero reading frame) (Fig 3)[102]. Thus, low expression of full-length SasG in the unmodified A2058 strain suggests the possibility of either a -1 ribosomal frameshift or a downstream stop codon readthrough, whereas a less translationally competent $m^6A2058$ ribosome further reduces SasG synthesis (Fig 4).

$m^6A$ modification of ribosomes not only reduces in vivo and in vitro fitness but also causes modest collateral sensitivity to a seemingly unrelated DNA gyrase inhibitor (ciprofloxacin), but curiously, not to nalidixic acid, another quinolone (Table 1). Strain USA300 is naturally resistant to ciprofloxacin due to mutations in the *gyrA* and *grlA* genes. We sequenced the regions and observed no nucleotide substitution in the two genes. Because $m^6A2058$ ribosomes reduce the expression of many cell surface components (Fig 3), it is possible that cell permeability increases upon the loss of cell-anchored proteins, resulting in a modest decrease in ciprofloxacin resistance.

The candidate genes affected by the $m^6A$ modification of A2058 in this work do not overlap with those found in a previous study [60]. For instance, ribosomal frameshift in *cidC* (*poxB*) and pausing in *tenA* were not detected in our Ribo-seq (S4C and S4D Fig). This discrepancy is largely because in the previous work, *ermC* was constitutively expressed on a multicopy plasmid under the control of a strong $P_{spac}$ promoter without the regulatory *ermCL* leader region. The layers of regulation via gene copy number, the promoter, and *ermCL* regions may be masked, as opposed to the use of a single copy native promoter-bearing *ermBL-ermB* operon in our study. Second, a cloning surrogate RN4220 strain of *S. aureus* was used in previous

comparative 2D gel-based proteomics, yielding only a handful of candidates [60]. RN4220 has reduced growth fitness and is known to carry loss-of-function mutations in many virulence factors and global regulators. It is widely recognized in the *S. aureus* community that RN4220 is not a suitable strain for studying antibiotic resistance and pathogenesis [103]. Our reconstituted MRSA strains are more similar to the natural MLS-resistant isolates because the observed phenotypes are highly reproducible in two independent sets of tagged strains (CdCl$_2$ and tetracycline markers), the catalytically dead mutant (*ermBL*$^{WT}$-*ermB*$^{Y103A}$) rescues all phenotypes to an *ermB*-minus level, and the hypermethylation variant (*ermBL*$^{R7stop}$-*ermB*$^{I75T/N100S}$) enhances the effects observed in a constitutively expressed strain (*ermBL*$^{R7stop}$-*ermB*$^{WT}$). Despite these experimental differences, our single-nucleotide-resolution Ribo-seq and previous proteomics unequivocally demonstrate that misregulated translation is the key feature of m$^6$A2058 ribosomes as part of the fitness cost to survive antibiotic stress.

## Materials and methods

### Ethics statement

All animal experiments were described previously [79] and approved by the Saint Louis University Institutional Animal Care and Use Committee (protocol 2640, PHS assurance number A-3225-01). Saint Louis University is an AAALAC-accredited institution and adheres to the standards set by the Animal Welfare Act and the NIH *Guide for the Care and Use of Laboratory Animals*.

### Strains, plasmids, chemicals, and growth conditions

The strains and plasmids used in this study are listed in S3 Table. Primers were purchased from IDT DNA and are listed in S4 Table. Strain JE2 is a community-associated methicillin-resistant *Staphylococcus aureus* (CA-MRSA) of USA300 lineage cured of all three native plasmids [104]. To construct *ermBL-ermB* containing JE2 strains, the operon was PCR amplified using genomic DNA from *S. aureus* CM05 (GenBank EF450709) as a template and cloned into the PstI and KpnI sites of pJC1111 or pJC1306 [57]. These suicide plasmids were integrated into the JE2 chromosome of strain RN9011 following standard protocol, and subsequently transferred to the JE2 strain by Φ11 phage transduction [57]. Quikchange site-directed mutagenesis (Agilent Genomics) was used to introduce *ermBL* or *ermB* mutations on the pJC1111 and pJC1306 derivatives followed by chromosomal integration as described above. ErmB (I75T/N100S) variant was isolated from a laboratory-directed evolution experiment by serial passaging of *S. aureus ermBL-ermB* with an increasing dose of erythromycin and clindamycin. Hyper-resistant colonies to both antibiotics were subjected to whole-genome sequencing. A pair of mutations in *ermB*, namely the *ermB*$^{I75T/N100S}$ allele was identified and combined with an *ermBL*$^{R7stop}$ background to obtain the constitutively expressed *ermBL*$^{R7stop}$- *ermB*$^{I75T/N100S}$ at the *att* site of the chromosome.

To induce the expression of *ermB* by sublethal dosage of macrolide, TSB cultures of OD$_{600}$~ 0.5 were split into halves, one portion was treated with 1 μg/mL of freshly made erythromycin and the half was supplemented with an equal volume of 70% ethanol (mock). Cells were grown at 37˚C for additional 60 min before harvesting for RNA or protein extraction.

Unless otherwise noted, *S. aureus* cells were grown aerobically at 37˚C in tryptic soy broth (TSB, BD Difco #211822). *E. coli* were grown in lysogeny broth (LB, BD Difco #244610) at a 5:1–10:1 tube- or flask-to-medium ratio with a 1:100 dilution of an overnight seed culture. When necessary, antibiotics were supplemented at the following final concentrations: erythromycin (5 μg/mL, 1 μg/mL when used as an inducer), chloramphenicol (10 μg/mL), kanamycin

(75 μg/mL), ampicillin (100 μg/mL), tetracycline (2.5 μg/mL) and cadmium chloride (0.15 mM). All chemicals were from Sigma-Aldrich unless otherwise noted.

## Measurement of minimum inhibitory concentration (MIC)

MIC values of various antibiotics were determined by E-test strips (Biomérieux or Liofilchem) on the Mueller Hinton agar (BD Difco #225250) plates following the manufacturer's instruction. MICs were recorded after 20 hr incubation at 37˚C.

## Detection of m⁶A methylation status by primer extension

Total *S. aureus* RNA was extracted using a modified hot-phenol-SDS method [105,106]. Two hundred fifty nanograms of total rRNA was used for primer extension as described previously [40] using the 5'-6-carboxyfluorescein (FAM)-labeled antisense oligonucleotides (S4 Table). DNA sequencing ladders were generated using a USB Thermo SEQ kit (Affymetrix) with *S. aureus* 23S rDNA as a template. The reverse transcribed products were resolved on 10% TBE-urea polyacrylamide sequencing gels and then scanned on a Typhoon 5 Imager (Cytiva).

## Reverse transcription-quantitative PCR (RT-qPCR)

Total RNA was extracted using a hot-phenol-SDS method [105,106] and further cleaned up with an RNeasy kit (Qiagen). DNA contaminants were removed using two successive digestions using Turbo DNase I (Ambion). RNA integrity was confirmed by a minimum accepted 1:1 intensity ratio of 23S:16S rRNA on a denaturing agarose gel stained with ethidium bromide. RT-qPCR procedures and internal reference gene (*polC*) were described previously [79], differences in mRNA levels were calculated using a published $2^{-\Delta\Delta CT}$ formula [107].

## In vitro competition assays

Overnight cultures of strains were diluted to an $OD_{600}$ of 0.18, and 20 μL of each strain was inoculated into 4 mL of TSB containing ~1.5×MIC of erythromycin (20 μg/mL for $ermBL^{WT}$-$ermB^{WT}$ strain). The 1:1 mixed culture were incubated with shaking at 37˚C and at each 24 hr, transfer to fresh TSB media at 1/1000 dilutions for 4 days. At 0, 48, and 96 hours, aliquots were removed, serial diluted and plated on plain TSB agar or TSB agar supplemented with the appropriate antibiotics for CFU enumeration.

## In vivo competitive coinfection experiments

Six-week-old female C57BL/J mice averaging 18±0.9 g (Jackson Laboratory) were intravenously injected with either 100 μL of phosphate-buffered saline (PBS), or 100 μL of a total of $2×10^5$ CFU of $ermBL^{WT}$-$ermB^{WT}$ and $ermBL^{R7stop}$-$ermB^{WT}$ mixture (1:1 ratio). On day 4 post-infection, the mice were euthanized. Mouse livers and kidney pairs were removed, homogenized in 1 mL of sterile PBS in a closed system tissue grinder (SKS Science), and dilution plated on TSB agar plates supplemented with appropriate antibiotics. CFU counts were recorded after 24 of incubation at 37˚C. Statistical significance was determined with one-way analyses of variance (ANOVAs). Tukey's multiple-comparison tests were performed after ANOVAs with GraphPad Prism 9 to analyze the differences in the bacterial burdens.

## Ribosome fractionation, Ribo-seq, and mRNA-seq

Crude ribosomes were isolated from *S. aureus* ($OD_{600}$~1.0) by cryo-milling methods [64,108] after harvesting cells by rapid filtration. Liquid nitrogen-frozen cell pellets were pulverized on Retch MM400 miller in lysis buffer [10 mM $MgCl_2$, 100 mM $NH_4Cl$, 20 mM Tris (pH 8.0),

0.1% IGEPAL CA-630 (Sigma-Aldrich), 0.4% Triton X-100 (Sigma-Aldrich), 100 U/mL RNase-free DNase I (Roche), 0.5 U/μL SUPERase•In (Ambion), 5 mM $CaCl_2$] [64, 99]. S7 micrococcal nuclease (Roche) was used to convert polysome into monosomes. Twenty-five absorbance units ($Abs_{260}$) of ribosomes were layered on a 10–40% sucrose gradient that was prepared on a BioComp Gradient Master. The samples were centrifuged at 210,000 ×$g$ at 4˚C in a SW41 rotor in a Beckman Coulter Optima XPN-100 ultracentrifuge for 3 hr. Fractionation was performed using a Brandel fractionation system equipped with a UA-6 UV detector. cDNA libraries of total mRNA and ribosome-protected mRNA fragments (RPFs) were prepared exactly as described [64,99]. Raw FastQ sequencing data were processed using a locally installed Galaxy platform [109]. The rRNA-less reads were aligned to the USA300 reference genome (GenBank CP000255) using Bowtie2. The alignment.map files were used as inputs for the published Python scripts [99,110] and was described in greater details in ref [99]. The normalized ribosome densities, measured as reads per million (RPM), were visualized in Mochi-View [111]. The gene expression levels (reads per kilobase per million mapped reads, RPKM) were calculated as follows: Read quantification was performed using featureCounts functionality [112]. Read counts were loaded and normalized into edgeR's Trimmed Mean of M values (TMM) algorithm [113]. Subsequent values were then converted to RPKMs. Differential expression analysis was performed using edgeR's test for differences between two groups of negative-binomial counts with an estimated dispersion value of .1. While RPKM is a widely accepted normalization for within-sample comparisons, RPM unit is useful for assessing overall expression levels between samples and has been used in all ChIP-seq, miRNA-seq and Ribo-seq analyses. In this work, RPKM will be used for measuring differential gene expression whereas RPM will be used for the ribosome density plot presentations. Translational efficiency (TE) was calculated as the relative number of ribosome footprints to mRNA-seq reads in $log_2$ ratios [114]. Processed and raw sequencing data were deposited in the NCBI GEO database with accession number GSE168265.

## Antibodies and Western blots

*S. aureus* cell pellets were homogenized with Lysing Matrix B (MP Biomedicals) in 25 mM Tris (pH 7.5) on a FastPrep-24 homogenizer (MP Biomedicals). Clarified lysates were recovered by spinning at 20,817×$g$ at room temperature for 5 min to remove cell debris. A total of 0.1–0.2 $Abs_{280}$ units of cell lysate were analyzed on 4–20% TGX SDS-PAGE gels (BioRad), or 4–12% Bis-Tris NuPAGE minigels (Invitrogen) and the proteins were transferred to a nitrocellulose membrane using a Trans-Blot Turbo system (BioRad). The membrane was stained with Ponceau red (Amresco #K793-500mL) to ensure equal loading, followed by immunoblotting using anti-ErmB (1/1,000 dilutions, a gift from Julian Rood [115]), anti-SdrE (1/1,000 dilutions, a gift from Dominique Missiakas), anti-SigB and anti-Asp23 (1/2,000 dilutions; 1/100 dilutions, gifts from Bischoff Markus), anti-YwlG (1/1,000 dilutions, [116]) anti-SasG (1/200, Abnova PAB16066), anti-mCherry (1/1,000 dilutions, Novus Biologicals NBP196752X), anti-YFP (1/500 dilutions, Roche 11814460001), anti-Nuc2 (1/500 dilutions, OriGene AP05314SU-N), anti-Goat Anti-SpA (1/2,000, OriGene AP05314SU-N), Donkey anti-Goat Alexa Fluor Plus 800 (1/10,000 dilutions, Invitrogen A32930), and HRP-conjugated anti-IgG secondary antibody (1/15,000 dilutions, Cytiva #NA9120). SuperSignal West Dura chemiluminescence substrate was used (Thermo Scientific #34075). Images were acquired using iBright FL1500 system (ThermoFisher).

## Dual fluorescence reporter assays

*S. aureus* cells carrying the reporter plasmids were grown at 37˚C in TSB until $OD_{600}$ = 0.4, 0.8, 1.4, and 2.0. At each cell density, cells were collected and washed with PBS for 3 times and

the final PBS suspensions of $OD_{600}$ optical density were transferred to a 96-well black side plate (Greiner Bio-One). The signals of mCherry, YFP, and $Abs_{600}$ were measured on a TECAN SPARK plate reader. The YFP/mCherrry ratio of the "error reporters" was normalized by the YFP/mCherry ratio of the "WT control reporter" in the $ermBL^{WT}$-$ermB^{WT}$ strain. Statistical significance was tested by unpaired two-tailed Student's t test in GraphPad Prism 9.0. Only p value less than 0.05 is considered statistically significant.

## Cell morphology analysis by fluorescence microscopy

When *S. aureus* TSB culture grown at 37˚C reached to $OD_{600} = 1$, one-hundred microliters of the cultures were incubated with 1 µg/ml of DNA dye Hoechst 33342 (Invitrogen) and 5 µg/mL of membrane dye FM4-64 (Invitrogen) at room temperature for 5 min. Two microliters of the sample were deposited directly onto slides and visualized at room temperature on a Nikon Eclipse 90i Widefield Fluorescent Microscope equipped with a Plan Fluor 100× Oil Iris DIC H/N2 objective and a Cytoviva filter box (excitation filter set for DAPI and mCherry). Images were acquired using a Photometrics CoolSNAP HQ2 camera with an exposure time of 1–3 seconds. Nis-Elements AR software (Nikon) was used for image capture. Images were analyzed by Fiji ImageJ software.

## Transmission Electron Microscopy (TEM)

*S. aureus* cells were grown to mid-log phase ($OD_{600}$~1.0) in TSB at 37˚C. Cells were harvested at 5,000 rpm for 10 min, washed once with PBS and the pellet was fixed 2.5% glutaraldehyde in 0.1 M sodium cacodylate buffer for 30 min. Cells were then embedded in 3% agarose and the solidified pads were kept in the fixture at 4˚C until use. It was then rinsed in cacodylate buffer and post fixed in 2% osmium tetroxide for one hour. The sample was then rinsed 3 times (10 min each) in $ddH_2O$, followed by a graded series of alcohol 50%, 75%, 95% and absolute alcohol twice. This was followed by propylene oxide and finally a 50/50 mixture of Epon 812 resin and Propylene Oxide overnight. The sample was embedded the following day in Polyscience Epon 812 resin in BEEM® capsules and cured in a 60˚C oven for 48 hours. Ultra-thin sections were obtained with a Leica Ultracut Uct microtome. The sections were stained in 5% uranyl acetate followed by lead citrate. Images were obtained using a FEI Tecnai Spirit G2 transmission electron microscope at the Center for Advanced Microscopy (CAM) Center at Northwestern University.

## Supporting information

**S1 Table. Removal of *ermBL* coding sequence from the *ermBL-ermB* operon renders low levels of macrolide resistance.** Minimum inhibitory concentration (MIC, in µg/mL) were determined by E-test on Muller Hinton Agar plates in duplicates per strain per antibiotic type. (PDF)

**S2 Table. Summary of Ribo-seq and mRNA-seq libraries reported in this study.** (PDF)

**S3 Table. Strains and plasmids.** (PDF)

**S4 Table. Oligonucleotides used in this study.** (PDF)

**S1 Dataset. Changes in translational efficiency (TE, RPF/mRNA) in A2058 vs m⁶A2058 ribosome containing strains.**
(XLSX)

**S1 Fig. Cell morphology and cell size were unaltered upon m⁶A modification of ribosomes.** **(A-C)** *S. aureus* strains (JE2 (*ermBL⁻ermB⁻*), *ermBL*^R7Stop^-*ermB*^WT^, *ermBL*^R7Stop^-*ermB*^Y103A^, *ermBL*^R7Stop^-*ermB*^I75T/N100S^) were grown overnight in TSB at 37˚C until $OD_{600} = 0.5$. Cells were stained with the membrane dye FM4-64 and the DNA dye Hoechst 33342 and visualized on an epifluorescence microscope. The images were acquired in an unbiased manner by using the multiple image alignment function in NIS Element software (Nikon). At least 50 cells distributed on >3 image frames were recorded. Scale bar = 2 μm. **(D)** Transmission electron microscopy (TEM) analysis detects no significant alteration in cell wall thickness and ultrastructure of *S. aureus* cells with and without an active ErmB. At least 35 cells distributed on 2 image frames were recorded. Representative single cells are shown. Scale bars represent 100 nm with 49,000× magnification.
(PDF)

**S2 Fig. Estimation of m⁶A2058 modification by primer extension.** Primer extension was performed with a fluorescently labeled primer P813 (S4 Table), 250 ng rRNA, and Superscript III reverse transcriptase in the presence of 1 mM dATP, dGTP, 0.25 mM dTTP and 1 mM ddCTP. The cDNA products were resolved on a 10% denaturing polyacrylamide gel. Reverse transcription is impeded by m⁶A2058. cDNA synthesis terminates at G2056 and serves as a normalization reference.
(PDF)

**S3 Fig. m⁶A modification does not affect ribosome assembly but increases the abundance of translationally inactive 100S ribosomes.** Ribosome sedimentation profiles (n = 3) of Ribo-seq samples prepared from *ermBL*^WT^-*ermB*^Y103A^ and *ermBL*^R7Stop^-*ermB*^WT^ strains were subjected to micrococcal nuclease (MNase) treatment or untreated and analyzed by 10–40% sucrose density gradient ultracentrifugation. The 100S ribosomes do not contain mRNA and are not collapsed into 70S monosomes like the polysomes, showing distinct peaks that are indistinguishable from the translating disomes on a sucrose gradient due to near identical masses. Peak height is indicated by the absorbance at 254 nm (y-axis). Each sedimentation profile contains twenty-five $Abs_{260}$ units of input crude ribosomes that were isolated from logarithmically grown *S. aureus* cells ($OD_{600}$~ 0.9–1.0).
(PDF)

**S4 Fig. Ribosome occupancy profile of genes with known frameshift event or ribosome stalling sites and candidate genes identified from previous study [60].** **(A)** Ribosome stalling peptide upstream of *ilv-leu* amino acid biosynthesis operon. Peptide sequence and stalling motif are highlighted in magenta. **(B)** +1 programmed frameshift in *prfB* gene. **(C)** Translational profile of *cidC* (formerly *poxB*). **(D)** Overall low ribosome occupancy (poor mRNA translation) in *tenA*. In panels C-D, regions highlighted in green indicate a frameshift site in *cidC* and a ribosome pausing site in *tenA*. These events were not observed in this Ribo-seq study likely due to the use of different *S. aureus* strains. A small RNA s900 is annotated upstream of *tenA*.
(PDF)

## Acknowledgments

We thank Victor Torres and Jiqiang Ling for generously sharing the *S. aureus* chromosomal integration system and plasmids, and Anna Liponska for technical assistance. Transposon

mutants were obtained through Network on Antimicrobial Resistance in *Staphylococcus aureus* (NARSA) for distribution by BEI Resources, NIAID, NIH.

## Author Contributions

**Conceptualization:** Kathryn E. Shields, Mee-Ngan F. Yap.

**Formal analysis:** Yongjun Tan, Dapeng Zhang, Mee-Ngan F. Yap.

**Funding acquisition:** Mee-Ngan F. Yap.

**Investigation:** Kathryn E. Shields, Mee-Ngan F. Yap.

**Methodology:** Kathryn E. Shields, David Ranava, Mee-Ngan F. Yap.

**Project administration:** Mee-Ngan F. Yap.

**Resources:** Yongjun Tan, Dapeng Zhang.

**Software:** Yongjun Tan, Dapeng Zhang.

**Supervision:** Mee-Ngan F. Yap.

**Visualization:** David Ranava.

**Writing – original draft:** Mee-Ngan F. Yap.

**Writing – review & editing:** Kathryn E. Shields, David Ranava, Yongjun Tan, Dapeng Zhang, Mee-Ngan F. Yap.

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
