## [Decision Letter · Decision Letter 0]

4 Dec 2023

Dear Dr Yap,

Thank you very much for submitting your manuscript "Epitranscriptional m6A modification of rRNA negatively impacts translation and host colonization in Staphylococcus aureus" for consideration at PLOS Pathogens. As with all papers reviewed by the journal, your manuscript was reviewed by members of the editorial board and by several independent reviewers. The reviewers appreciated the attention to an important topic. Based on the reviews, we are likely to accept this manuscript for publication, providing that you modify the manuscript according to the review recommendations.

Sincerely,

Anne Jamet

Guest Editor

PLOS Pathogens

Marcel Behr

Section Editor

PLOS Pathogens

Kasturi Haldar

Editor-in-Chief

PLOS Pathogens

orcid.org/0000-0001-5065-158X

Michael Malim

Editor-in-Chief

PLOS Pathogens

orcid.org/0000-0002-7699-2064

Reviewer Comments (if any, and for reference):

Reviewer's Responses to Questions

**Part I - Summary**

Reviewer #1: The authors of the paper entitled “Epitranscriptional m6A modification of rRNA negatively impacts translation and host colonization in Staphylococcus aureus” aimed to understand the bacterial physiological and gene expression effects of a rRNA modification that generates macrolide resistance in S. aureus. It is known from previous publications, including a couple from the senior corresponding author of this work, that changes in ribosome components, especially ribosomal proteins that generate antibiotic resistance can affect gene expression in bacteria as well as growth fitness. This work shows that methylation of nucleotides at rRNA that confer resistance to macrolides also affect gene expression and growth fitness. The authors used a series of ermBL and ermB mutant genes to enhance the presence of methylation at the 23S rRNA A2058 nucleotide of S. aureus. Bacteria carrying such methylation showed alteration in transcription and translation of several genes, corresponding as well with the incapacity of the cells to compete in vitro and in vivo with non modified cells. Despite that the main question of the work has been addressed before, the application of new methods (ribosome profiling) gives a new perspective to this important scientific area. The document is well written and the experimental designs are well constructed. The work opens avenues for further experimentation to understand the mechanism involved in these observations.

Reviewer #2: In their manuscript, Shields and coauthors studied the impact of 23S ribosomal methylation upon translation efficacy, colonization and infection. They focus on the methylation of A2058 on the 23S rRNA subunit which is responsible for antibiotic resistance to MLS. The authors show that the ermBL leader sequence is critical for the regulation of MLS resistance. Then, they did competition experiments using strains that have different markers (Tet or CcCl2) to show that hypermethylation of A2058 is detrimental for in vitro fitness or infection in a murine model in the absence of an antibiotic stress. They perform Ribo-seq and mRNA seq and described that many genes related to virulence, metabolism and information processing are downregulated in hypermethylated strain although a substantial number of genes is upregulated. At the end of their study, they used a nice dual-fluorescence reporter to determine translational output and accuracy in hypermethylated strain. The story is well-written, somehow short and provide additional information that goes beyond antibiotic, although the real impact of such study is not easy to appreciate. The story is well constructed but there are things that needs to be improved or better explained.

Line 140: There is not many information regarding the ermBLR7STOP-ermBI75T/N100S variant. In the experimental procedures, details regarding on how the authors obtained this variant is very limited. Then, I don’t understand why the authors did not construct an ErmBLWT-ermBI75T/N100S variant. In table 1, MIC are the same compared with ErmBLWT-ermBWT variant, except for SOL. Using the ErmBLWT-ermBI75T/N100S variant would have clearly show that it can serve as a positive control for hypermethylation.

Line 142: The authors mentioned an “extremely low basal level of methylation”. It seems that there is actually nothing, or very similar that that of the negative control Y103A.

The competitive fitness experiment is interesting, but why the authors did not try to test infection in murine model under erythromycin condition to verify if in vitro results are fully reproducible?

Line 188: The authors used epifluorescence to study bacterial morphology. However, electron microscopy and measurement of cell wall thickness may have been more appropriate.

The authors do not sufficiently describe the section related to RPF and mRNAseq. They do not provide any numbers regarding the actual number of genes that are downregulated and upregulated. It seems that there are more genes upregulated that downregulated. This information is really important. Although I can understand that they do not have an explanation regarding this increase, they cannot just state that many virulence genes are downregulated, which tend to send a message to there an overall decrease in gene expression. Additionally, why not doing a DEG analysis using DESeq2 for mRNAs?

Figure 4: Why there is almost nothing on the lane ermBL—ermB- for Nuc2?

Line 255: Based on results in WT, one could have imagined the the level of Yfp should be lowered in all condition. Please comment.

**Part II – Major Issues: Key Experiments Required for Acceptance**

Reviewer #1: To enhance the presentation of this work, this reviewer suggest the following:

Figure 1D. The primer extension signals above the A2028 methylation signal are not consistent with expectations. These weak signals are shown at ribosomes from cells without ermB cassettes; they are also shown in cells with the ermBLR7stop, but not with ermBL wild type under either + or - erythromycin. Are the amounts of rRNA similar per each sample? Would be great if the authors show somewhat that each lane reaction used the same amount of rRNA.

Also, could the authors determine % of methylated rRNA by using their primer extension assays? Comparison between their data and previous observations (Gupta et al. Nat Commun. 2013;4:1984.) seems to be essential for their manuscript.

Regarding stability of ribosomes or ribosome synthesis: the ribosome sedimentation assays shown at Fig S2 could be used to calculate the amount of ribosomes per cell?

Figure 1E. Is there any way that the authors could concentrate the protein sample to get a stronger signal?

Figure 2B. I suspect the authors might have done in vivo experiments with erythromycin as well. If such, would it be significant to show them as well? It might correlate with the in vitro data.

Lines 208-210: The authors wrote:“Local accumulation of ribosomes at specific codon positions of mRNA followed by sparse downstream ribosome density is indicative of ribosome stalling. Such a signature was not observed in either strain.” This reviewer is wondering if there are known gene sequences in S. aureus that produced such arrest signatures seen for example in other bacteria genes such as secM, tnaC, speFL, in E. coli, that could be used as positive controls of their method.

Also, the authors might want to address the previous observations about the gene poxB (Gupta et al. Nat Commun. 2013;4:1984.)

Figure 5B, could the authors suggest an explanation of why methylation of A2058 would affect the wild type YFP/mcherry ratio? Since it is a long peptide, it seems that the ribosome modification is either reducing YFP production or increasing mCherry production. This reviewer was expecting to see the same ratio if both ORFs were affected equally. Could it be a ribosome arrest prior to reaching the YFP ORF?

Also, for the case of WT and TAG: could the authors suggest that reduction seen in this figure could be instead of “slight”, 50 and 25% reduction respectively?

Reviewer #2: Line 140: There is not many information regarding the ermBLR7STOP-ermBI75T/N100S variant. In the experimental procedures, details regarding on how the authors obtained this variant is very limited. Then, I don’t understand why the authors did not construct an ErmBLWT-ermBI75T/N100S variant. In table 1, MIC are the same compared with ErmBLWT-ermBWT variant, except for SOL. Using the ErmBLWT-ermBI75T/N100S variant would have clearly show that it can serve as a positive control for hypermethylation.

Line 188: The authors used epifluorescence to study bacterial morphology. However, electron microscopy and measurement of cell wall thickness may have been more appropriate.

The authors do not sufficiently describe the section related to RPF and mRNAseq. They do not provide any numbers regarding the actual number of genes that are downregulated and upregulated. It seems that there are more genes upregulated that downregulated. This information is really important. Although I can understand that they do not have an explanation regarding this increase, they cannot just state that many virulence genes are downregulated, which tend to send a message to there an overall decrease in gene expression. Additionally, why not doing a DEG analysis using DESeq2 for mRNAs?

**Part III – Minor Issues: Editorial and Data Presentation Modifications**

Reviewer #1: Line 271, consider changing the following: “changes are due to an indirect” to “changes could be due to an indirect”

Line 287, change “dysregulated” to “dysregulated”

Table 1, could be better for the viewer to change “nd” to “-”. The table seems busy.

Reviewer #2: Line 142: The authors mentioned an “extremely low basal level of methylation”. It seems that there is actually nothing, or very similar that that of the negative control Y103A.

The competitive fitness experiment is interesting, but why the authors did not try to test infection in murine model under erythromycin condition to verify if in vitro results are fully reproducible?

Figure 4: Why there is almost nothing on the lane ermBL—ermB- for Nuc2?

Line 255: Based on results in WT, one could have imagined the the level of Yfp should be lowered in all condition. Please comment.

PLOS authors have the option to publish the peer review history of their article (what does this mean?). If published, this will include your full peer review and any attached files.

Reviewer #1: No

Reviewer #2: No

Figure Files:

Data Requirements:

Reproducibility:

References:

---

## [Editor Report · Decision Letter 1]

12 Jan 2024

Dear Dr Yap,

We are pleased to inform you that your manuscript 'Epitranscriptional m6A modification of rRNA negatively impacts translation and host colonization in Staphylococcus aureus' has been provisionally accepted for publication in PLOS Pathogens.

Best regards,

Helena Ingrid Boshoff

Section Editor

PLOS Pathogens

Marcel Behr

Academic Editor

PLOS Pathogens

Kasturi Haldar

Editor-in-Chief

PLOS Pathogens

orcid.org/0000-0001-5065-158X

Michael Malim

Editor-in-Chief

PLOS Pathogens

orcid.org/0000-0002-7699-2064

The authors have fully addressed all reviewers' concerns and the additional experiments have further strengthened this work.
---

## [Editor Report · Acceptance letter]

18 Jan 2024

Dear Dr Yap,

We are delighted to inform you that your manuscript, "Epitranscriptional m6A modification of rRNA negatively impacts translation and host colonization in Staphylococcus aureus," has been formally accepted for publication in PLOS Pathogens.

Best regards,

Michael Malim

Editor-in-Chief

PLOS Pathogens

orcid.org/0000-0002-7699-2064